# A Molecular-Wide and Electron Density-Based Approach in Exploring Chemical Reactivity and Explicit Dimethyl Sulfoxide (DMSO) Solvent Molecule Effects in the Proline Catalyzed Aldol Reaction

**DOI:** 10.3390/molecules27030962

**Published:** 2022-01-31

**Authors:** Ignacy Cukrowski, George Dhimba, Darren L. Riley

**Affiliations:** Department of Chemistry, Faculty of Natural and Agricultural Sciences, University of Pretoria, Lynnwood Road, Pretoria 0002, South Africa; u16402902@tuks.co.za (G.D.); darren.riley@up.ac.za (D.L.R.)

**Keywords:** proline catalyzed aldol reaction, reaction mechanism, explicit solvent effects, DMSO, REP-FAMSEC method, chemical reactivity, forces driving a chemical change

## Abstract

Modelling of the proline (**1**) catalyzed aldol reaction (with acetone **2**) in the presence of an explicit molecule of dimethyl sulfoxide (DMSO) (**3**) has showed that **3** is a major player in the aldol reaction as it plays a double role. Through strong interactions with **1** and acetone **2**, it leads to a significant increase of energy barriers at transition states (TS) for the lowest energy conformer **1a** of proline. Just the opposite holds for the higher energy conformer **1b**. Both the ‘inhibitor’ and ‘catalyst’ mode of activity of DMSO eliminates **1a** as a catalyst at the very beginning of the process and promotes the chemical reactivity, hence catalytic ability of **1b**. Modelling using a Molecular-Wide and Electron Density-based concept of Chemical Bonding (MOWED-CB) and the Reaction Energy Profile–Fragment Attributed Molecular System Energy Change (REP-FAMSEC) protocol has shown that, due to strong intermolecular interactions, the HN-C-COOH (of **1**), CO (of **2**), and SO (of **3**) fragments drive a chemical change throughout the catalytic reaction. We strongly advocate exploring the pre-organization of molecules from initially formed complexes, through local minima to the best structures suited for a catalytic process. In this regard, a unique combination of MOWED-CB with REP-FAMSEC provides an invaluable insight on the potential success of a catalytic process, or reaction mechanism in general. The protocol reported herein is suitable for explaining classical reaction energy profiles computed for many synthetic processes.

## 1. Introduction

Organocatalysts have historically been utilized to catalyze a range of non-asymmetrical organic transformations, most notably Knoevenagel condensations, esterifications, Baylis-Hillman reactions and Stetter reactions [1,2]. Attempts to develop organocatalyzed asymmetrical transformations led to the development of the Hajos-Parrish-Eder-Sauer-Wiechert reaction in the 1970s [3,4]. However, following this breakthrough, the development in the field remained largely limited until the late 1990s [2]. In the past two decades, the increasing demand for pure and optically active compounds in chemical industries and academia and a growing drive for greener metal-free catalytic processes has prompted a renaissance in the field of asymmetric organocatalysis [2,5,6,7]. As a result, one can now access vast libraries of organocatalysts that can be utilized for a multitude of different chemical transformations. Notably, the use of proline and related analogues in asymmetric synthesis has continued to see development, becoming one of the most widely utilized classes of organocatalysts [8,9,10,11].

One of the more important transformations catalyzed by proline is the economic direct aldol reaction, wherein a C–C bond is formed between simple carbonyls [12,13,14], allowing access to enantiomerically rich intermediates. Proline has been shown to exhibit a similar effect to that of type 1 aldolase enzymes [12,15], wherein the C–C bond formation is preceded by the formation of a key enamine intermediate [12,16,17]. Proline’s efficiency, as an organocatalyst in the aldol reaction, has in turn led to several investigations aimed at elucidating the mechanistic details of the transformation [17,18,19,20].

As pertaining to the proline catalyzed aldol reaction, Ajitha and Suresh proposed in 2011 [21], based on a density functional theory (DFT) study, that the lowest energy conformer (LEC) of (*S*) proline [22] was inactive with its reaction pathway not proceeding beyond an initial proton transfer step. In 2019, we confirmed these findings using the REP-FAMSEC technique [23] (Reaction Energy Profile–Fragment Attributed Molecular System Energy Change). This approach goes beyond the classical use of reaction energy profiles by identifying fragments of a molecular system and quantifying their contributions in either driving, facilitating, or inhibiting the progression of a reaction [23,24].

In previously reported computational modelling studies of the aldol reaction [18,20,21,23,25,26,27], an implicit solvent model has been explicitly used. This is despite the fact that:The aldol reaction is reported to proceed better when performed in dimethyl sulfoxide (DMSO) [12,15,28].Incorporating explicit solvent molecules produced more reliable computed activation free energy barriers in modelling of numerous reaction mechanisms [29,30].Discrete solvent molecules can capture solvent dynamics [31] and may play a significant role in chemical reactions that implicit solvation models fail to capture [32,33].

Hence, the major aim of this work is to explore and explain the mechanistic roles played by explicit DMSO solvent molecules and the carboxylic group of proline (called a co-catalyst [12]) in the proline catalyzed aldol reaction. Our special focus is on the initial step described by List et al. [12] as ‘the nucleophilic attack of the amino group’ that leads to a CN-bond formation. List et al. [12] hypothesized that ‘This co-catalyst may facilitate each individual step of the mechanism, including the nucleophilic attack of the amino group’ but were not able to support it. This initial step is of critical importance because it fixes the ketone-coupling partner, e.g., acetone, through the newly formed CN-bond with proline and this is a pre-requisite for consecutive steps to proceed successfully. To achieve our aims we took advantage of the recently reported Molecular-Wide and Electron Density-based concept of Chemical Bonding (MOWED-CB) [34] and the REP-FAMSEC [23,24] protocol as only through exploring entire molecular environments can one identify and understand reaction forces leading to a chemical change, e.g., intramolecular re-arrangement or breaking/forming of classical chemical bonds.

## 2. Computational Details

All calculations were performed in Gaussian 09 Rev. D01 [35] at the RB3LYP/6-311++G(d,p) level of theory with Grimme’s [36] empirical correction for dispersion (GD3). It was established that, using CCSD(T) as a benchmark, B3LYP-GD3, M06, and M06-2X (with the 6-31+G(d,p) basis set) give a reasonable, within a few kcal/mol, electronic and Gibbs free energies when modelling reaction mechanisms [37]. All calculations were performed using a hybrid implicit–explicit solvation model with DMSO as a solvent employing the implicit default solvation model–coordinates and energies of all structures discussed in this work are provided in Appendix A. Frequency calculations were performed for the optimized local, global, and transition state (TS) structures. Zero and one imaginary frequency were obtained for minimum energy (local and global) and TS structures, respectively. The lowest energy pathway connecting a given transition state with the two associated energy minima (intrinsic reaction coordinate–IRC) was calculated to verify each transition state. Topological calculations were performed in AIMAll (ver. 19.02.13) [38] using B3LYP-generated wavefunctions as IQA energy terms, and interaction energies in particular, were found to be highly comparable to those obtained at the CCSD/BBC1 level [39]. To discover the lowest energy 3-, 4-, and 5-molecular complexes with one, two, and three DMSO solvent molecules, respectively, were subjected to a conformational search performed in Spartan [40]. Each molecular complex contained a single molecule of proline and acetone. The atom numbering of individual molecules is shown in Figure 1. Numbering of atoms in complexes always starts with proline, hence its numbering remains the same throughout, followed by acetone (i.e., C1 becomes C18 and so on) followed by molecules of DMSO (e.g., S1 becomes S28).

## 3. Basic Concepts of REP-FAMSEC Method Applicable to This Work

A detailed account of the concept and potential applicability of the REP-FAMSEC method have been described previously [23]. Hence, to aid the interpretation of generated energy trends, only basic ideas and expressions relevant to molecular systems of interest to this work will be presented and explained.

We consider a molecular system as a 3D assembly of any number of atoms that are mathematically treated on an equal footing using an IQA [41,42] (Interacting Quantum Atoms) electronic energy partitioning scheme. All atoms of a molecular system interact with each other and, as we have previously explained in detail [23], the interatomic interactions are influenced by far more (often by more than an order of magnitude) than just atomic energies when a chemical event occurs. Monitoring, quantifying, and interpreting of mainly, but not exclusively, changes in interaction energies is the fundamental concept incorporated in the REP-FAMSEC method. The main inputs come from a diatomic interaction energy term, $E_{\mathrm{int}}^{A,B}$, and its components (a Coulomb or classical term, $V_{\mathrm{cl}}^{A,B}$, and an exchange-correlation term, $V_{\mathrm{XC}}^{A,B}$, which is commonly used as a measure of the degree of covalent contribution). In accordance with the MOWED-CB [34] concept, these terms are computed for each unique atom-pair A,B in a molecular system regardless of (i) the internuclear distance between them, d(A,B), or whether atoms are considered as being chemically bonded. One must stress that even the smallest displacement of one (or more) atoms within a molecular system will always change all $E_{\mathrm{int}}^{A,B}$ values computed for a full set of unique atom-pairs. Naturally, the significance of the interaction energy change, $\Delta E_{\mathrm{int}}^{A,B}$, depends on the extent of the atoms’ displacement due to a chemical event and is largest for atom-pairs containing displaced atoms and their immediate (closest) neighbors.

From a classical perspective, atoms of a molecular system might belong to a single molecule (due to a network of covalent bonds) or several molecules. In the MOWED-CB world, however, a molecule or poly-molecule system is considered as a constellation of atoms linked by a network of interactions. In a larger molecular system, typically, only a few atoms will experience a significant diatomic interaction energy variation on a chemical event, e.g., more than ±10 kcal/mol. These atoms are considered in the REP-FAMSEC approach as most responsible for a chemical change and their interactions are seen as forces driving a chemical change through a specific reaction mechanism; hence, these atoms’ strongest (decisive) interactions are monitored and analyzed along the reaction coordinates. It is then appropriate and convenient to consider the selected atoms as a molecular *n*-atom fragment *G* of a molecular system. Each molecule may have a set of most ‘influential’ atoms, when a reaction mechanism is considered, and they will be treated as separate fragments. Moreover, there might be more than one molecular fragment in a single molecule. Typically, changes in interaction energies are monitored in a stepwise fashion along the reaction coordinates as Δ*E*_int_ = ^fin^*E*_int_ − ^ini^*E*_int_, where fin and ini refer to the final (after a chemical change) and initial (prior to a chemical change) structure (or 3D placement of atoms) of a molecular system. The Δ*E*_int_ term might provide invaluable insight at any stage of a process under investigation, e.g.,:Formation of a poly-molecular complex from separate molecule—from this, one can learn how and why molecules arrange themselves relative to each other, and which atoms drive such arrangement.Inclusion of a solvent molecule to a poly-molecular complex—does this impact relative placement of molecules in the complex, what is the solvent molecule’s preferred site and why.Can molecules re-arrange themselves ‘freely’ within a complex and which atoms drive the molecules to attain their lowest, or global minimum structure.What drives molecules to better pre-organization required for subsequent bond formation or breaking, etc.

To gain a full picture and understand the reaction mechanism, we will analyze many interaction energy terms, such as intra- and intermolecular, covalent and long-distance non-covalent interactions, all of them computed either for a single molecule or for grouped molecules. This requires a specific, purposeful grouping of diatomic interaction energy $E_{\mathrm{int}}^{A,B}$ terms. As a consequence, numerous and not commonly encountered expressions quantifying such energy terms will be introduced; for convenience they are placed together with descriptions in Appendix B at the end of the manuscript. 

## 4. Results and Discussion

### 4.1. Exploring the Number of DMSO Molecules in an Explicit Solvation Model

When implementing an explicit solvation model, in which discrete molecule/s of solvent are included in the computational modelling, the number of solvent molecules to be added and their positioning relative to the solute molecules is still a subject of debate [43]. We decided to pay special attention to two aspects in our preliminary investigations, namely:The minimum number of explicit DMSO solvent molecules needed to strike a balance between the computational cost and insights derived knowing that the computational time and a number of intermolecular interactions increase exponentially with a number of atoms in a molecular system. We decided to limit the number of DMSO solvent molecules to three at most and use a smaller basis set in our preliminary studies, namely 6-31+G(d,p), rather than 6-311++G(d,p) employed in this work.Specific properties of our molecular system in terms of leading intermolecular interactions between proline **1** and acetone **2**. The input structures for conformational searches in Spartan had a relative arrangement of **1** and **2** such that the intermolecular H-bonding O16–H17···O19 was preserved. This is because our recent findings [23] revealed that the transfer of H17 from proline **1** to O19 of acetone **2** must take place as it largely facilitates the CN-bond formation occurring between N13 (in **1**) and C18 (in **2**).

#### 4.1.1. Geometrical Considerations

Numerous structures of the 3-MC, 4-MC, and 5-MC (MC = molecular complex) discovered by Spartan were optimized in Gaussian. Only pre-organized structures for the concurrent transfer of H17 (from **1**) to O19 (of **2**) and a CN-bond formation between N13 (of **1**) and C18 (of **2**) were selected for further studies. Complexes containing the LEC of **1** are shown in Figure 2, where dashed lines indicate common geometrical features observed in all complexes; values represent average interatomic distances obtained for all complexes. Notably, the N13,C18 and H17,O19 atom-pairs are also linked by dashed lines. Importantly, the interatomic distances do not vary dramatically with an increase in the number of DMSO molecules as indicated by rather small standard deviations.

Comparable general geometrical features, but with shorter interatomic distances and smaller standard deviations, are also observed for complexes containing the higher energy conformer (HEC, **1b**) of proline **1**—Figure 3. As an example, the distance between atoms destined to form a new bond in the first step of the catalytic process d(N13,C18) of 3.3 ± 0.3 Å (with **1a**) and 2.62 ± 0.08 Å (with **1b**) was found, on average, in 3-, 4-, and 5-MCs.

#### 4.1.2. Leading Diatomic Interactions

Two critical intermolecular di-atomic interactions must be considered. They involve N13,C18 and H17,O19 atom-pairs, as two new bonds are formed during the initial stage, N13–C18 and O19–H17 [23]. These interactions are much stronger in **1b** complexes and hardly vary in strength with the addition of DMSO molecules—Figure 4. To this effect, the interaction energy $E_{\mathrm{int}}^{N13,C18}$ = –143.1 ± 1.7 stays nearly unchanged in complexes containing **1b** and, on average, is stronger by over –25 kcal/mol when compared with molecular complexes containing **1a**. From structures in Figure 2 and Figure 3, it is clear that only one DMSO molecule is involved in a strong diatomic intermolecular interaction with H5 of proline **1**. This interaction is highly comparable in all complexes; on average $E_{\mathrm{int}}^{H5,O37}$ of –102.0 ± 2.0 and –97.2 ± 3.0 kcal/mol was found, respectively, in **1a**- and **1b**-containing complexes—see Figure 4. Due to very strong diatomic interactions between **1** and **2**, as well as **1** and a DMSO molecule **3**, the 3-MC feature is seen in all complexes, regardless of the number of solvent molecules. One might say that the 3-MC is a ‘fixed’ and quite rigid structure that is solvated by adding more DMSO molecules. Rigidity of the 3-MCs is significantly strengthened by additional and very strong attractive and repulsive diatomic interactions. To illustrate this, see data included in Table 1 where the strongest intermolecular diatomic interactions between **1b** and acetone **2** are presented. Small standard deviations in computed interaction energies show compellingly that if solvent molecules play any role in this catalytic process, then it must be mainly due to the DMSO molecule that forms an ‘inseparable’ 3-MC by being anchored to H5 of **1**; additional DMSO molecules in 4- and 5-MCs do not impact interactions between **1** and **2** significantly. 

#### 4.1.3. The Energy Barrier Computed for the First Step of the Catalytic Process

Conclusions arrived at from the analyses employing geometric data (Section 4.1.1) and diatomic intermolecular interactions (Section 4.1.2) strongly indicate that investigating the 3-MC with a single DMSO molecule should be sufficient to provide the sought after answer of whether DMSO is just a medium or if it does play a role in a catalytic process. In search of further support for the above assumption, we decided to model the first step of the catalytic process in the presence of 3-, 4-, and 5-MCs; results obtained are included in Table 2.

Firstly, and as one would expect, energies computed at transition states do vary slightly with the number of DMSO molecules as there are many degrees of freedom in placing DMSO molecules relative to each other as well as relative to molecules **1** and **2**. However, the network of strongest interactions remains unchanged—Figure 4. Focusing on *E*_ZPVE_ as an example, the energy barrier at the TS for **1a**-containing complexes is 7.8 kcal/mol for the 3-MC, slightly smaller for the 4-MC (7 kcal/mol) and somewhat larger for the 5-MC, 10.0 kcal/mol; this translates to 8.3 ± 1.6 kcal/mol on average for 3-, 4-, and 5-MCs. A much smaller and even more consistent value of 1.6 ± 0.6 kcal/mol was obtained at the TS for **1b**-containing complexes—Table 2, part (b).

Considering the 3- and 4-MCs, the energy barriers are consistently lower for **1b**-containing complexes, respectively, by 5.6 and 6.0 (for *E*_ZPVE_), 5.4 and 6.2 (for *H*), and 6.1 and 5.7 (for *G*) kcal/mol. This shows that the addition of a second DMSO molecule does not have a decisive impact on the energy barriers at the TSs. 

Therefore, accounting for all the findings discussed above, we concluded that a single molecule of DMSO that interacts directly and very strongly with **1** might play a significant role; hence, our further studies are restricted to 3-MCs.

### 4.2. Impact of a DMSO Solvent Molecule on the Reaction Energy Profile

The reaction energy profiles obtained for the 2- and 3-MCs (up to the second proton transfer) are presented in Figure 5.

Small letters represent data obtained at each consecutive step along the reaction coordinates for 2-MCs, i.e., in the absence of a DMSO molecule. Relevant data obtained in the presence of an explicit DMSO molecule are marked with capital letters. Letters **a**/**A** and **b**/**B** refer to the molecular systems containing the LEC (**1a**) and HEC (**1b**), respectively. To illustrate the impact of a DMSO solvent molecule, the energy trends are presented relative to the combined energies of isolated reactants of the 2- and 3-MCs. Full sets of energies obtained for each stationary point and relevant structures are included in Appendix A. Only the most important observations and conclusions pertaining to the first and second proton transfer follow.

#### 4.2.1. Concomitant First Proton Transfer and the C–N Bond Formation Step

Looking at the relevant free energy data for the first H-transfer (Figure 5b,d), it is clear that the DMSO molecule had no significant impact on the free energy barrier Δ*G*^‡^ at the **5-TS** stage; relative to the isolated molecules, data obtained for the 2- and 3-MCs is comparable.

A very different picture is seen for trends in the *E*_ZPVE_ values (Figure 5a,c) as the presence of a DMSO molecule decreased the electronic energies of both molecular systems (MSs)–the black line trace (3-MC) is well below the red one. Importantly, the values computed at transition states **5A**/**B_TS** for the 3-MCs are also below the energies of isolated reactants, whereas the relevant values at **5a**/**b_TS** for 2MCs (without the DMSO explicit solvent molecule) are positive and hence, less favorable. Moreover, the energy barriers Δ*E*^‡^_ZPVE_ (i.e., the difference in the electronic energy between the TS and the GMS) decreased in the presence of a DMSO molecule by nearly six (for the HEC) and two (for the LEC) kcal/mol. This means that a DMSO molecule is minimizing the energy barrier at transition states regardless of whether or not the MS contains **1a** or **1b**, but the energy has been decreased by far more in the case of the HEC (**1b**). As a result, small Δ*E*^‡^_ZPVE_ values of about 3.1 and 9.4 kcal/mol in the case of HEC- and LEC-containing MSs, respectively, are observed in the presence of the explicit DMSO solvent molecule.

Energy levels computed for the products of the first H-proton transfer deserve special attention. Considering **5a_eq** and **5b_eq** (i.e., in the absence of an explicit DMSO molecule, 2-MCs), the red line traces for *E*_ZPVE_ and *G* seen in Figure 5 reveal that the products of the first proton transfer are a few kcal/mol higher in energy than the global minimum structures (GMS) of 2-molecular complexes, **4a**/**b_GMSs**. A reversal of this unfavorable trend is observed in the presence of an explicit DMSO molecule, but only in the case of the HEC (**1b**). Notably and specifically, for the reaction energy profile obtained for *E*_ZPVE_ (Figure 5c, black trace), the energy of **5B_GMS** is lower by –6.2 kcal/mol relative to **4B_GMS**. The opposite is true for the product containing **1a**, i.e., **5A_eq**. Its energy is higher than **4A_GMS** by +1.4 kcal/mol as shown in Figure 5a, black trace.

It has been established [21,23] that structural interconversion from **1a** to **1b** does not require a large energy barrier. This means that the two conformers are always present in a reaction vessel. Combining this knowledge with the data shown in Figure 5a,c implies that the LEC of proline is not catalytically active at all, as it will not even become involved in the first H-transfer stage. This is because, when both global minimum structures of the 3-MCs, **4A_GMS** and **4B_GMS**, are present in a solution, the system will always follow a downwards change in its energy that leads to the most stable **5****B_GMS** state containing the higher energy conformer of proline. Note that a system with the LEC must climb the hill of *E*_ZPVE_ to reach the **5A_eq** state and this is not only higher in energy than the starting point, i.e., the **4A_GMS** 3-MC, but also higher in energy, by +3.3 kcal/mol, than the **5B_GMS** of 3-MC with the HEC of proline.

This clearly shows that the mechanism through the higher energy conformer **1b** becomes even more favorable in the presence of the explicit DMSO solvent molecule. In other words, the presence of a DMSO molecule not only decreases the energy barrier but also differentiates between the two conformers of proline. This conclusion is further and strongly supported by the free energies of products **5** of the first H-transfer/CN-bond formation step (Figure 5b,d). Relative to the reactants in the form of the global minimum 3-MCs, **4A_GMS** and **4B_GMS**, the change in the free energy Δ*G* of +4.9 and –1.6 kcal/mol is observed, respectively, for the first stage products, **5A_eq** and **5B_GMS**. Furthermore, the value of *G* computed for **5B_GMS** with the HEC is lower, by –2.7 kcal/mol, than that obtained for the **5A_eq**, i.e., a hypothetical product of the first stage involving the LEC of proline.

Our computational modelling confirmed that the explicit DMSO molecule is not directly involved in a chemical change at the first stage. At the same time, from the trends shown in Figure 5a,c it is obvious that DMSO acts as a catalyst by decreasing the transition state energies. From that it follows that to understand the role played by a DMSO molecule **3**, one must initially examine intermolecular interactions between **3** and molecules of proline **1** and acetone **2**, as these interactions must have caused a decrease in the *E_ZPVE_* values and resulted in significantly smaller energy barriers for the first H-transfer.

It was then of importance to investigate a change in the intermolecular interactions at the two transition states, **5A_TS** and **5B_TS** relative to the reactive 3-MCs. The term Einterint3,1,2 = Einterint3,1 + Einterint3,2 was calculated. It quantifies the total intermolecular interaction energy of DMSO, Einterint3,1,2, which is made of interactions between atoms of the DMSO molecule **3** and atoms of proline **1**, Einterint3,1, and atoms of acetone **2**, Einterint3,2. We found that at the TSs, the intermolecular interactions between a DMSO molecule and:Proline plus acetone strengthened significantly as Einterint3,1,2 changed favorably (became more negative) by –55.3 and –75.7 kcal/mol for **5A_TS** and **5B_TS**, respectively. Hence, exactly the same set of diatomic intermolecular interactions, between **3** and **1** plus between **3** and **2**, became stronger, by –20.4 kcal/mol, in the case of the **1b**-containing MS.Acetone strengthened more, by –7.1 kcal/mol, in the case of the LEC-containing MS; the change in the Einterint3,2 term of –14.7 (at **5A_TS**) and –7.6 (at **5B_TS**) kcal/mol was obtained.Proline strengthened a lot and in favor of **1b**-containing MS by –27.5 kcal/mol; the change in the Einterint3,1 term of –40.6 (**5A_TS**) and –68.1 (**5B_TS**) kcal/mol was obtained.

From the above, it is clear that the larger increase in the strength (by –27.5 kcal/mol) of the interactions between the DMSO molecule and the HEC of proline at the **5B_TS** must have led to a more significant decrease in the energy barrier discovered for the **1b**-containing MS.

#### 4.2.2. Second Proton Transfer

Although we have shown that the LEC cannot be catalytically active beyond the first proton transfer, we will analyze the impact of a DMSO explicit molecule for the two proline conformers for illustrative purposes. Relative to the **1a**-containing 2-MC, the energy barriers between **5A_eq** and **6A_TS** in Figure 5a,c became even larger in the presence of DMSO. The energy barriers increased by +6.4 and +5.3 kcal/mol for *E*_ZPVE_ and *G*, respectively. Consequently, the free energy barrier at the **6A_TS** reached an insurmountable Δ*G*^‡^ value of 46 kcal/mol. Moreover, the product of the second proton transfer **6A_eq** is much higher in energy than that of the first proton transfer, **5A_eq**, regardless of whether *E*_ZPVE_ or *G* is considered. It is then clear that the reaction energy profile obtained for the LEC prohibits any reaction progress when starting from reactants, but it also shows that the reverse process, from **6A_eq** toward reactants would be, if permitted, a spontaneous process.

On the other hand, the presence of a DMSO molecule has a facilitating impact on the reaction progress involving the HEC. Figure 5c,d shows that:The energy barriers Δ*E*^‡^_ZPVE_ and Δ*G*^‡^ (from **5B_GMS** to **6B_TS**) are very low, just a few kcal/mol for both energy terms.The energy difference between a transition state **6B_TS** and the product of the second proton transfer **6B_eq** decreased slightly in the presence of DMSO.

These two findings, however, do not matter at all. What really matters is the energy difference between **6B_eq** and **4B_GMS**. The most abundant **1b**-containing 3-MC of the reactants (**4B_GMS**) after overcoming two negligible energy barriers becomes the product of the second proton transfer **6B_eq** that finds itself in a energetically favorable position. This is because its *E*_ZPVE_ is below the energy of **4B_GMS** by –2 kcal/mol. Notably, the equivalent energy difference obtained for 2-MCs (without DMSO) was found to be +2.6 kcal/mol. Hence, the *E*_ZPVE_ energy term for **6b_eq** is slightly less favorable when compared with the starting materials, i.e., **4b_GMS**. Moreover, the **1b**-containing molecular systems become more stable in the presence of a DMSO molecule, by –4.6 kcal/mol relative to 2-MC.

A very interesting picture is observed in Figure 5d, where **4B_GMS** (3-MC with DMSO) is significantly higher in energy than **4b_GMS**. However and importantly, the increase in energy caused by a DMSO molecule must have a beneficial effect because the energy difference between the product of the second proton transfer **6b-eq** and **4b_GMS** of +7.4 kcal/mol (red line trace) was nearly nullified to +0.3 kcal/mol for the 3-MC (black line trace linking **4B_GMS** and **6B_eq**). This means that in the presence of DMSO, there is essentially no backward driving force, from **6B_eq** to the starting material **4B_GMS** and hence the reaction can proceed forward ‘unopposed’.

### 4.3. Molecular Interactions Driving a Chemical Change

Reaction energy profiles, such as those in Figure 5, are very useful computational tools used by physical organic chemists in predicting the most likely and rejecting the most unlikely reaction mechanisms for a synthetic process of interest. A conclusion is solely made based on the activation energy barriers at transition states. Often such predictions coincide with experimental data (when available), but do not provide any deeper explanation related to chemical reactivity or the role played by solvent or non-covalent interactions. To build a knowledge base needed to understand reaction mechanisms and forces driving a chemical change on a fundamental level, down to atomic scale, one must fully explore all possible modes of interactions. This is also very clear from trends in Figure 5, where variations in energy levels can be attributed only to interactions between all three molecules.

For a chemical reaction to proceed towards a desired product, a poly-molecular system must pre-organize itself such that atoms destined to form new bonds face each other and are as close to each other as possible. Without any doubt, the same leading forces, i.e., interactions between specific molecular fragments and atoms will drive (or obstruct) such a re-arrangement as well as a chemical reaction of interest. Hence, to explore the forces leading to a chemical change, it is of importance to discover the processes leading to the global minimum structures (GMSs) of a molecular system, as they constitute the majority of species present in a solution. In the best-case scenario, the GMS is pre-organized for the required chemical change, but this is not always the case. Hence, to estimate an energy barrier at the TS, the energy required from the GMS to best pre-organized structure would have to be accounted for.

Separate sets of five 3-MCs were prepared for **1a** and **1b** conformers of proline using a protocol described in Appendix A. Each set had an input structure, three local minimum (LM) structures and a GMS–**1a**-containing 3-MCs are shown in Appendix A whereas relevant complexes containing **1b** are shown in Figure 6. Each set contains the pre-organized structure necessary to model the first stage of this catalytic process as well as intermediate stationary points through which a molecular system must proceed in order to reach the best structural arrangement prior climbing the energy barrier of the TS. A full set of energies (i.e., electronic (*E*), zero-point vibrational energy corrected electronic energy (*E*_ZPVE_), enthalpy (*H*), and the Gibbs free energy (*G*)) computed for all structures seen in Figure 6 and Figure 7 is included in Appendix A. It is worthwhile to note that the **4A_GMS** (Appendix A) is not a well pre-organized structure due to an ‘incorrect’ placement of **2** relative to **1a**. The **4A_LM-3** structure (Appendix A) that is slightly higher in energy is best pre-organized for the new CN-bond formation and proton transfer, H17 from **1a** to O19 of **2**. Considering **1b**-containing structures, the **4B_GMS** (Figure 6) is also the best pre-organized structure.

In the sections that follow, we will explore in some detail the forces driving a chemical change and the role played by a DMSO solvent molecule.

#### 4.3.1. Changes in the Electronic Energy, Gibbs Free Energy, and the Total Molecular System Interaction Energy

Figure 7 shows that the formation of the 3-MC is energetically favorable as, relative to separate components, Δ*E*_ZPVE_ decreased by –9.3 and –12.6 kcal/mol for the **4A_inp** and **4B_inp** systems, respectively. It is also clear that a classical analysis, based on trends in the Δ*E*/Δ*G* values, cannot provide any significant information on what drives molecular systems towards a GMS and the subsequent first stage of a catalytic process. This is because throughout the **4_inp** → **4_GMS** rearrangement, the Δ*E*_ZPVE_ and Δ*G* values remain rather small, they are nearly constant (relative to the **4_inp** structures, *E*_ZPVE_ decreased just by –2.5 kcal/mol in the **4_GMSs**) and are highly comparable for both molecular systems.

Remarkably, the $\Delta E_{\mathrm{int}}^{\mathrm{MS}}$ terms (they incorporate all intramolecular and intermolecular interaction energy contributions computed for entire molecular systems) are nearly an order of magnitude more significant than the Δ*E*_ZPVE_ values. Molecules became instantly involved in strong interactions and $\Delta E_{\mathrm{int}}^{\mathrm{MS}}$ = –72.9 kcal/mol obtained for **4B_inp** is about –10 kcal/mol more significant than the value computed for the **4A_inp** with **1a**, i.e., the lowest energy conformer of proline. Whereas electronic and Gibbs free energies appear to be ‘insensitive’ to large structural re-arrangement from the input to the global minimum structures, the total interaction energies show significant variation and a clear trend. These interactions became stronger in both **4B_GMSs**. However, the $\Delta E_{\mathrm{int}}^{\mathrm{MS}}$ term of –118.2 kcal/mol shows that interactions in **4B_GMS** are, relative to **4A_GMS**, stronger by over –40 kcal/mol. Trends seen n Figure 7 lead to the obvious conclusion that:The interactions drive the 3-MCs formation in the first place.Strengthening of interactions is a leading force in the **4_inp** → **4_GMS** structural re-arrangement.To gain a deeper understanding of a chemical change, one must explore different modes of interactions.Much stronger interactions found for **4B_GMS** already point to the **1b**-containing complex for which a smaller energy barrier at TS should be expected, and this agrees with trends seen in Figure 5 very well.

#### 4.3.2. Total Intramolecular and Intermolecular Interaction Energies Computed for Individual Molecules

To understand the role a DMSO solvent molecule plays, it is necessary to decompose the total molecular system interaction energy to components linked with individual molecule, i.e., to compute the $E_{\mathrm{int}}^{\mathrm{mol}}$ energy term. This term is comprised of contributions derived from covalently bonded atoms making a skeleton of a molecule, non-covalent long-distance (L−D) intramolecular interactions, as well as intermolecular interactions between atoms of a selected molecule and atoms of all remaining molecules. This approach follows a recent molecular-wide and electron density-based concept of chemical bonding [34] and the REP-FAMSEC method [23] that provides all necessary tools to implement it in studying reaction mechanisms or, in general, exploring forces driving a chemical change.

Considering the LEC of proline **1a**, a change in molecular interaction energy $\Delta E_{\mathrm{int}}^{\mathrm{mol}}$ is calculated as $\Delta E_{\mathrm{int}}^{1a}$ = ΔintraEint1a + Einterint1a,2 + Einterint1a,3. The first term, ΔintraEint1a, accounts for the change in the intramolecular (intra-proline) interactions on the complex formation from **1a**, **2**, and **3**, whereas the latter two terms account for intermolecular interaction energies between atoms of **1a** and all atoms of the other two molecules in the 3-MCs, i.e., **2** (acetone) and **3** (a DMSO solvent molecule). From Figure 8, we found that:Quite surprisingly, **1a** is the only molecule for which interactions weakened as the $\Delta E_{\mathrm{int}}^{1a}$ term changed from –74.5 (in **4A_inp**) to –61.1 kcal/mol (in **4A_GMS**).Molecular interactions computed for **1b** are not only much stronger when compared with **1a**, but they strengthened immensely, from $\Delta E_{\mathrm{int}}^{1b}$ of –105.9 (in **4B_inp**) to –151.6 (in **4B_GMS**) kcal/mol. This means that molecular interactions computed for **4B_GMS** are more than twice as strong as those obtained for **4A_GMS**.Only trends computed for individual molecules of **1b**-containing complexes follow the trend for $\Delta E_{\mathrm{int}}^{\mathrm{MS}}$ throughout.Looking at data obtained for a DMSO molecule in **1a**-containing complexes, its interactions strengthened more than that found for (i) a molecule of acetone and (ii) a DMSO molecule in **1b**-containing complexes.

Using common sense and chemical intuition, one would expect a good catalyst (here proline) to be involved in the strongest interactions in a given molecular system and the solvent molecule to be involved in the weakest interactions. This is indeed observed in Figure 8, but only for the 3-MCs with **1b**.

#### 4.3.3. Intermolecular Interaction Energies between Individual Molecules

It is reasonable to assume that stronger intermolecular interactions computed for the entire molecular system EinterintMS should bring molecules closer to each other and this, in turn, must facilitate a catalytic process. We found that trends in EinterintMS follow the trends in $\Delta E_{\mathrm{int}}^{\mathrm{MS}}$ (combined, intermolecular, and intramolecular interactions), but the EinterintMS values are more significant (i.e., more negative)—see Appendix A. Notably, intermolecular interactions EinterintMS of –144.2 kcal/mol in **4B_GMS** are much stronger, by –61 kcal/mol, than those in **4A_GMS**; this strongly points at **1b**, the HEC of proline, as being a better catalyst and is in a full agreement with reaction energy profiles seen in Figure 5.

To gain insight into the role played by individual molecules, we analyzed intermolecular interactions between each unique molecule-pair trends obtained, which are shown in Figure 9. Let us start from **1** and **2**, as one would assume that proline and acetone should play the most decisive and leading role in driving the reaction forward as they are destined to form a new CN-bond. Considering the lowest energy conformer of proline, there are at least two important and entirely surprising observations one can make, namely:The strongest intermolecular interactions are not between **1a** and acetone **2**, but between **1a** and a DMSO solvent molecule **3**; this is not what one would like to see at all.Significant weakening of interactions between **1a** and **2** (from –34.0 to –15.3 kcal/mol) took place on the transition from **4A_inp** → **4A_GMS**—see red trace. This shows that ability to drive a catalytic process weakens when **1a** spontaneously reaches its global minimum structure.

Notably, interactions between **1b** and **2** (the Einterint1b,2 term) strengthened from –52.6 (in **4B_inp**) to –84.5 (in **4B_GMS**) kcal/mol. Clearly, the higher energy conformer of proline **1b** must be a better and maybe the only active catalyst as the latter value is over five times more significant than that obtained for **1a**,**2** interactions computed for the **4A_LM-3** and **4A_GMS**. This finding is again in perfect harmony with data shown in Figure 5.

Furthermore, interactions between acetone **2** and a DMSO molecule **3** are much stronger and are strengthened even further (by –31.1 kcal/mol) in the case of 3-MCs with **1a**, whereas they hardly changed in complexes with **1b**. Intuitively, one might speculate that the strong interactions between acetone **2** and a solvent molecule **3** need to be weakened before a reaction between **1a** and **2** can proceed and this, potentially, might result in an increase in the associated energy barrier at a TS.

Finally, intermolecular interactions between **1** and **3** are strong in both systems and, relative to **4A_GMS**, they are about –20 kcal/mol stronger in **4B_GMS** but, importantly, still much weaker than interactions between the major players, **1b** and **2**. This suggests that the solvent molecule must play a significant role at this stage of the catalytic process.

We also analyzed interactions of single molecule with the remaining two molecules of the molecular systems and noted that:Relative to the input structure **4A_inp**, the Einterint1a,2,3 = Einterint1a,2 + Einterint1a,3 term increased (interactions weakened) by about 19 kcal/mol in **4a_GMS**, whereas the ΔinterEint1b,2,3 term became more negative, i.e., interactions strengthened by about –47 kcal/mol when in **4B_GMS**.The combined intermolecular interactions between **1b** and {**2**+**3**} of –139.6 kcal/mol in **4B_GMS** are 2.7 times stronger than the interactions between **1a** and {**2**+**3**} in **4A_GMS**.

### 4.4. Molecular Fragments Driving a Chemical Change

The above shows that all molecules interact strongly with each other and, in general, the trends discussed above illustrate the importance of interactions in modelling reaction mechanisms. It is obvious, however, that not all atoms of molecules forming a molecular system play significant and comparable roles. Hence, to identify atoms driving a chemical change, we change the focus from the molecular to atomic level.

Atoms with the largest positive (C14, S28, C18, H17, H5) and negative (O37, O15, O19, O16, N13) net atomic charges (Appendix A) are also involved in the strongest attractive (e.g., $E_{\mathrm{int}}^{C14,O37}$ = –171 kcal/mol) and repulsive (e.g., $E_{\mathrm{int}}^{C14,S28}$ = +148 kcal/mol) intermolecular interactions (Appendix A). Such strong and mainly electrostatic interactions drive the arrangement of molecules that subsequently leads to bonds breaking and forming. It became obvious that it is not a single atom-pair that drives the process, as typically adopted in a classical approach. We grouped selected atoms into molecular fragments and a full set of trends generated for interaction energies between molecular fragments 𝒜 = {C18,O19} in acetone, 𝒟 = {S28,O37} in DMSO and 𝒫 = {C1,C4,H5,N13,C14,O15,O16,H17} in proline is shown in Appendix A. As an example, and for the purpose of illustration, Figure 10 shows large differences in inter-fragment interaction energies computed for **4A** and **4B** molecular systems and they are clearly in favor of the latter. One would expect that the leading fragments of molecules that are to form new bonds (**1** and **2**) should be involved in the strong(est) interactions. This is indeed found in **4B_GMS** with $E_{\mathrm{int}}^{P,A}$ = –62.8 kcal/mol, but surprisingly $E_{\mathrm{int}}^{P,A}$ ~ 0 kcal/mol is observed for **4A_GMS**—see Figure 10a. Note also that $E_{\mathrm{int}}^{P,A}$ = –6.6 kcal/mol is seen for the pre-organized **4A_LM-3** structure.

The combined intermolecular interactions of 𝒫 with (𝒜 plus 𝒟) shown in Figure 10b reveal a strong attraction between atoms of 𝒫 in proline (**1a** and **1b**) and atoms of 𝒟 in DMSO as values of $E_{\mathrm{int}}^{P,\left(A,D\right)}$ are more negative than $E_{\mathrm{int}}^{P,A}$ in Figure 10a. This finding is of importance as it shows that the DMSO solvent molecule assists in attaining the most desired 3D placement of **1** and **2** for subsequent bond formation. On the other hand, interactions between the 𝒜 and 𝒟 fragments are very weak (0.0 and +1.4 kcal/mol in **4A_LM-3** and **4B_GMS**, respectively see Appendix A) and this is exactly what a synthetic chemist would like to see as the process of bonding between proline and acetone will not require additional energy for breaking interactions between acetone **2** and DMSO **3**. Remarkably, relative to **4A_LM-3**, interactions between the three fragments in Figure 10b are three times stronger in **4B_GMS** and this must have a huge and facilitating impact on the bond formation between N13 and C18 as well as O19 and H17; this correlates well with the lowering of the activation energy at **4B_TS** when **1b** is involved.

### 4.5. Importance of N13,C18 and H17,O19 Atom-Pairs

The N13,C18 and H17,O19 atom-pairs are destined to form two new bonds at the first stage of this catalytic process. Hence, their diatomic interactions must be seen as navigating a chemical change even though they are not the strongest (selected intermolecular diatomic interaction energies are presented in Appendix A). Trends in intermolecular diatomic interaction energies $E_{\mathrm{int}}^{N13,C18}$ and $E_{\mathrm{int}}^{H17,O19}$ shown in Figure 11 constitute additional and critical support for (i) selecting best pre-organized **4A_LM-3**, rather than commonly used global minimum structures and (ii) predicting a lower energy barrier for **4B_GMS** as indeed found in this study (Figure 5). Notably, the $E_{\mathrm{int}}^{H17,O19}$ of –150.6 kcal/mol is stronger (more attractive) than $E_{\mathrm{int}}^{N13,C18}$ by –20.2 kcal/mol in the **4B_GMS**. It is then clear that the interaction between H17 and O19 is in the driving seat when a chemical change occurs. The opposite trend is observed for **4A_LM-3**, where a difference of –22 kcal/mol in favor of $E_{\mathrm{int}}^{N13,C18}$ was found. This reveals that, in principle, the same catalytic process might proceed by a somewhat different mechanism when different conformers are involved. The first stage is mainly driven by the H17,O19 and N13,C18 atom-pair, respectively, in the system with the higher (**1b**) and lowest (**1a**) energy conformer of proline.

A full set of interaction energies between either N13 or H17 of proline **1** and atoms of either A or D is shown in Appendix A (see also Appendix A). Let us follow a classical approach where interactions between two atoms are commonly considered, in this case between N13 and C18 [15,20,26]. This interaction is highly attractive with $E_{\mathrm{int}}^{N13,C18}$ = –111.0 kcal/mol in **4A_LM-3**, and even more so with –130.4 kcal/mol in **4B_GMS**. To follow such an approach, however, one must provide scientifically sound answers to the following two questions:The attractive interaction between N13 and C18 is far from being the strongest between atoms of proline **1** and acetone **2**; so, why is only this interaction considered?There are also very strong repulsive interactions between atoms of proline **1** and acetone **2**; why are they not considered at all?

Clearly, there is no justification for such a simplistic and orthodox approach. So let us follow the molecular-wide and electron density-based concept of chemical bonding and do some simple mathematics related to the main actors. This means that a broader spectrum of interactions must be accounted for when the role played by N13 and H17 of proline **1** is considered.

Summing up the interaction energies between N13 and the remaining major players, i.e., atoms of A (C18 and O19 of acetone **2**) and 𝒟 (S28 and O37 of DMSO **3**), we obtained +24.4 (for **4B_GMS**) and +36.7 (for **4A_LM-3**) kcal/mol. In both cases, a large repulsive interaction was computed that would prevent the CN-bond formation. This is rather unexpected and a surprising finding, but one must realize that, for example, the attractive interaction between N13 and C18 in **4A_LM-3** (–111 kcal/mol) is counteracted by more significant repulsive interactions, $E_{\mathrm{int}}^{N13,O19}$ = +123.9 kcal/mol, between N13 and O19 (the atom C18 is bonded to). Considering **4B_GMS**, we obtained the attractive interaction between N13 and C18 (of –130.4 kcal/mol) that counteracts, but only barely, the repulsive interaction between N13 and O19 (of +129.4 kcal/mol).

Following the above approach used for the interactions involving N13, we summed up interaction energies between H17 and the remaining major players. A very different picture emerged as we obtained –68.4 (in **4B_GMS**) and –30.8 (in **4A_LM-3**) kcal/mol. Overall, large attractive interaction energies were obtained that promote a proton transfer from proline **1** to acetone **2**. Importantly, only in the case of **4B_GMS**, the attractive interactions involving H17 (of –68.4 kcal/mol) compensated repulsive interactions involving N13 (of +24.4 kcal/mol). Hence, using major players as a predictive tool, the reaction should not proceed via the LEC of proline **1** at all from the very first step as it was also predicted from the trends in Figure 5.

### 4.6. Special Role Played by H17 in ***1b***-Containing Complexes

The above data suggests that the H17 atom of a carboxylic group in proline plays an important, or even special, role in this catalytic process. To gain more insight, we analyzed additional trends that were computed for the **1b**-containing complexes—see Figure 12. Starting from a classical, 2-atom approach, the H17···O19 interaction is highly attractive from the very beginning (already starting from the **4B-inp** structure–Figure 12a) and persists to dominate the diatomic N13···C18 interaction throughout the entire process of structural rearrangement, from **4B_inp** to **4B_GMS**. Clearly, H17 attracts the in-coming acetone molecule much more than N13.

As mentioned already, it is imperative to also account for the obstructive (repulsive) interactions between N13 and O19 as well as H17 and C18 when **2** is approaching **1b**. To achieve this, we computed interaction energies between either N13 or H17 and the molecular fragment 𝒜 = {C18,O19} of acetone; trends obtained (Figure 12a) strongly point to H17 as a driver of a chemical change. This is because N13 is initially opposing the approaching acetone—see the top trend for N13, in Figure 12a where approximately +20 kcal/mol, hence highly repulsive interaction for the **inp** and **LM-1** complexes, is observed. In contrast, a very strong attractive force acts between H17 and 𝒜 throughout the **4B_inp** → **4B_GMS** rearrangement—see the H17, trace in Figure 12a. The picture does not change much when 4-atom fragments 𝒫2 (with N13) and 𝒫6 (with H17) instead of individual atoms N13 and H17 are considered (Figure 12b). Regardless of whether interactions with a single or both atoms of 𝒜 are considered, it is the 𝒫6 fragment with H17 that attracts **2** much more than 𝒫2 with N13.

### 4.7. Forces in the Driving Seat of a Simulated CN-Bond Formation

Considering the catalytic reaction between proline and acetone, the CN-bond formation is undoubtedly the most important initial chemical change for a classical organic chemist. Hence, we decided to monitor interaction energies between selected atoms and fragments throughout the process of a simulated CN-bond formation. Data obtained from scanning reaction coordinates, starting from d(N13,C18) = 2.6573 Å in **4B_GMS**, through a transition state TS at d(N13,C18) = 1.9527 Å and up to 0.2 Å beyond the TS, is shown in Figure 13. Focusing on just diatomic interactions (see Figure 13a), the H17,O19 atom-pair continues to be in the driving seat up to d(N13,C18) = 2.257 Å, where the two atom-pairs experience similar diatomic attraction of about –165 kcal/mol. Beyond this point, the interaction energy between N13 and C18 starts to dominate and is –16 kcal/mol stronger than the interaction between H17 and O19 at the transition state. Interestingly, well before the CN-bond is formed, i.e., at d(N13,C18) = 1.7573 Å, H17 permanently leaves proline and forms a new bond with O19 of acetone—see the insert in Figure 13a.

It is reasonable to assume that H17 would move to acetone even earlier if not for being restrained by a strong attraction to O16. To produce more realistic trends, one must consider a wider molecular environment. The interaction energies between atoms in question, N13 or H17, and the molecular fragment 𝒜 (see Figure 13b) show dominance of H17 over N13, in terms of attracting acetone, in the entire region of the reaction coordinates scan. Additional trends obtained for the interactions between molecular fragments 𝒫2 and 𝒫6 and either atoms of a molecular fragment 𝒜 or the entire 𝒜 are presented in Appendix A and they fully support what we observe in Figure 13b.

From this it follows that the first step’s name as a CN-bond formation, or as it was coined many years ago ‘the nucleophilic attack of the amino group’, is somewhat misleading. In our opinion and based on evidence provided in this work, a better description of the first step could be ‘a first proton transfer/CN-bond formation’, as this correctly reflects the sequence of chemical changes taking place during this step. Furthermore, the first step is the result of the attraction between two atom-pairs that guide **2** in its approach to **1** and there is no way to separate, either experimentally or theoretically, these chemical events, which take place nearly simultaneously. Due to the leading role played by H17 (it is involved in largely dominating attractive interaction with O19 of acetone) and for brevity we refer to this step as the first proton transfer also because it is a part of a multi-step proton transfer catalytic process involving proline. This is the dominating catalytic activity of a proton of a carboxylic group that not only facilitates the CN-bond formation, but makes ‘the nucleophilic attack of the amino group’ possible. Moreover, during the second H-transfer stage, H5 rebuilds the carboxylic COOH group of proline through an intramolecular mechanism, moving across from N13 to O16 [23]. The presence of the COOH functionality is a pre-requisite for the third consecutive H-transfer step during which water is eliminated by H5 extracting the O19H17 group from the acetone back-bone.

## 5. Conclusions

Considering a solvent just as a solubilizing medium for reactants and products might be, as this work demonstrates, far from the full picture. Through the use of the REP-FAMSEC [23,24] approach and the concept of MOWED-CB [34] (MOWED-CB = Molecular-Wide and Electron Density based concept of Chemical Bonding), the role of DMSO is revealed to extend well beyond that of a simple spectator. We demonstrated that DMSO, although not involved in bond forming/breaking or as a classical catalyst, is a major player in the aldol reaction. Modelling of the proline (**1**) catalyzed aldol reaction (with acetone **2**) in the presence of an explicit molecule of DMSO (**3**) has revealed that, due to strong intermolecular interactions, stable 3-molecular complexes (3-MCs) are instantly formed. Importantly, it is the HN-C-COOH (of **1**), CO (of **2**), and SO (of **3**) fragments that lead the 3-MCs to the global minimum structures (GMSs) and, as a matter of fact, they must be seen as driving a chemical change throughout the catalytic reaction. Essentially, a DMSO molecule plays a double role as it 1) leads to the elimination of the lowest energy conformer **1a** (LEC) as a catalyst at the very beginning of the process, namely at the first H-transfer/CN-bond formation and 2) acts as a facilitator by promoting the catalytic ability of the higher energy conformer (HEC) of proline **1b**.

We found that exploring (in detail) the process of structural rearrangement starting from separate molecules, through intermediate local minimum structures to the global minimum structure (GMS) of 3-MCs provides an initial and invaluable insight on either a success or failure of a potential catalytic process/mechanism. At the same time, molecular fragments of each molecule that drive the process can be identified and their role can be quantified. We found that the failure of **1a** as a catalyst was in the making from the very beginning as the total interactions, intramolecular and intermolecular in GMS of the 3-MCs, weakened for **1a**, but significantly strengthened for **1b**. Importantly, they became 2.5 times stronger when compared with the same set of interactions computed for **1a**-containing GMS of the 3-MC. Moreover and opposite to what one expects from a potential catalyst, the interaction between **1a** and acetone **2** became weaker. We found the opposite for the **1b**···**2** intermolecular interactions that became over five times stronger than **1a**···**2** in the GMS of the 3-MCs. We discovered a similar trend for the intermolecular interactions between **1** and {**2**+**3**}, i.e., the remaining molecules of the 3-MC, acetone **2** and the explicit DMSO solvent molecule **3**. The combined intermolecular interactions between the higher energy conformer **1b** and acetone **2** plus a DMSO molecule **3**, i.e., **1b**···{**2**+**3**}, are 2.7 times stronger than **1a**···{**2**+**3**} in the global minimum structures of the 3-MCs.

Mechanistically, although the N-atom of **1** clearly acts as a harbor for the acetone molecule **2** in the aldol reaction, the actual CN-bond formation is shown to be preceded by a proton transfer from the carboxylic acid group to the oxygen of **2**. Thereafter, a second proton transfer takes place, from the resulting quaternized nitrogen, to rebuild the carboxylic acid group that is necessary for the next proton involvement that leads to water elimination. Once again, DMSO is shown to play key and diverse roles in shaping reaction energy profiles by influencing the energies for the transformation of the LEC and HEC-based molecular systems. Importantly, this effect is by far more favorable for the **1b**-containing molecular system as the product of the first H-transfer is lower in energy than the proceeding GMS, but only in the case of **1b**-containing 3-MC. This energetic observation and above-mentioned interaction-based preferences for **1b** when coupled with the fact that **1a** and **1b** can readily interconvert with minimal energy in solution shows that the 3MC derived from the LEC can be considered catalytically inactive and essentially “cut-off” at the first proton transfer with the reaction preferring to proceed via **1b**. Furthermore, although unlikely, if the mechanism involving **1a** is able to proceed beyond the first proton transfer, it is for all intents and purposes completely cut off at the second H-transfer due to an insurmountably energy barrier. Once again, the dynamics of the DMSO molecule feature prominently facilitating stronger interactions between H5 of proline **1** and O37 of DMSO **3**, hindering (but only for **1a**) the required intramolecular transfer of H5 to O16 within proline **1**.

Interestingly, List previously reported that not only the CN-bond formation, but most likely the entire catalytic process is to be mediated through a series of proton transfers involving the CO_2_H group of proline **1** [12]. It is notable that the findings reported herein are in a manner pre-meditated by List [12] who suggested that:clearly both the pyrrolidine ring and the carboxylate are essential for efficient catalysis to occur and, based on experimental data, noted thatafter screening several solvents, we found anhydrous DMSO at room temperature to be the most suitable condition regarding reaction times and enantioselectivity.

Although our focus is on the proline catalyzed aldol reaction, the general approach described in this work that involves detailed analysis of the processes leading to the pre-organized complexes formation using the REP-FAMSEC method and MOWED-CB [34] concept will be applicable to many synthetic processes. This is because the interaction energies vary much more, typically by an order of magnitude, than commonly computed energy terms (*E*, *H* or *G*) used in drawing reaction energy profiles. Importantly, the REP-FAMSEC-based approach allows one to uncover subtle underlying relationships between atoms/fragments/molecules and how they influence each other; these are not easily predictable, if at all, using only classical approaches and chemical intuition. By investigating any and on purpose selected modes of interactions, one can fully explore and explain not only the role played by a solvent molecule(s), but also complex mechanistic processes can be rationalized in terms of molecular fragments driving the chemical change. From that, one can also identify catalytically (in)active conformers as well as suggest possible additional functionalities needed to improve a synthetic/catalytic process.

## Figures and Tables

**Figure 1 molecules-27-00962-f001:**
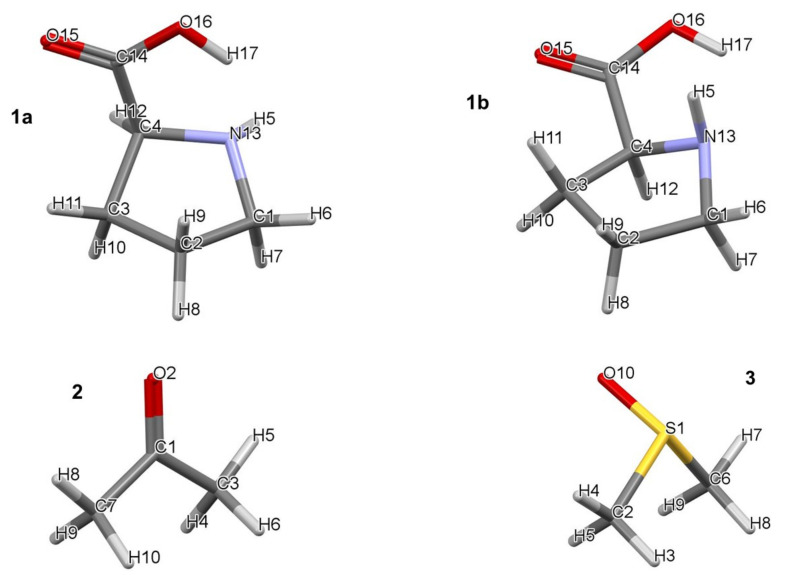
Numbering of atoms in lowest (**1a**) and higher (**1b**) energy conformers of proline (**1**), acetone (**2**), and DMSO (**3**).

**Figure 2 molecules-27-00962-f002:**
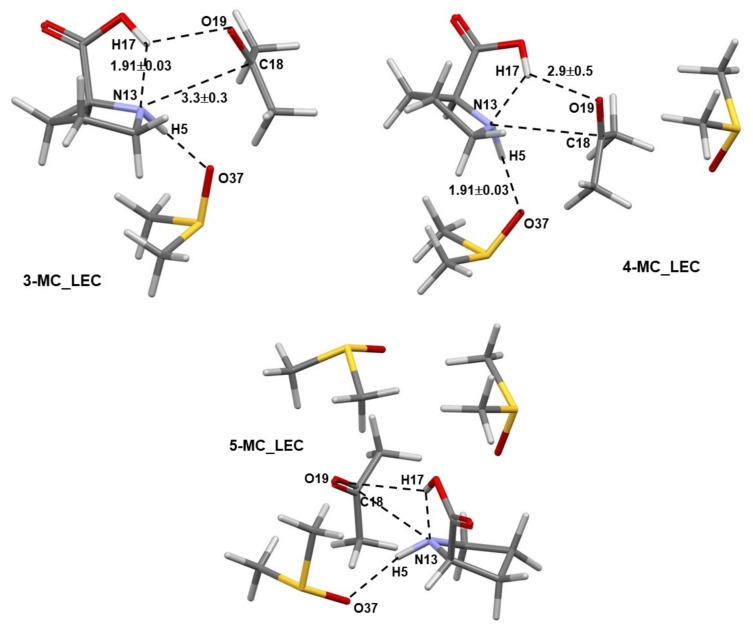
Selected averaged geometric distances with standard deviations (in Å) in the lowest energy and pre-organized (for the first step of the proline-catalyzed reaction with acetone) 3-MC, 4-MC, and 5-MC (found from conformation search in Spartan and energy-optimized in Gaussian) containing the lowest energy conformer (the LEC, **1a**) of proline **1**.

**Figure 3 molecules-27-00962-f003:**
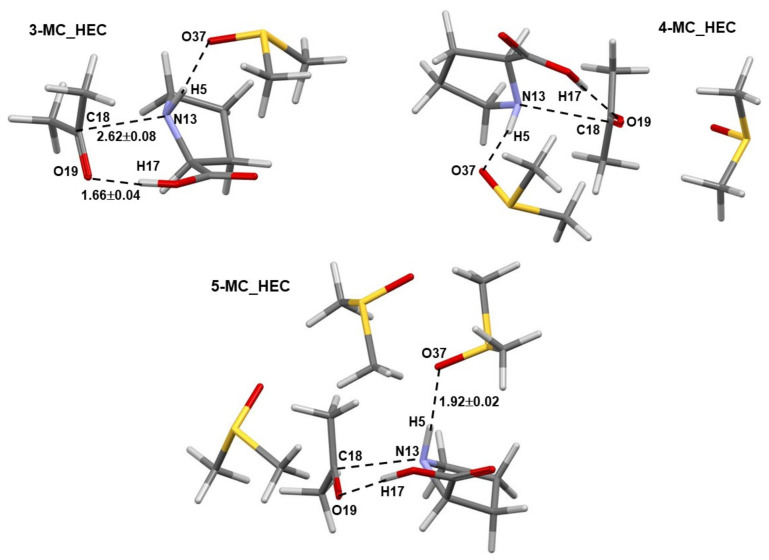
Selected averaged geometric distances with standard deviations (in Å) in the lowest energy and pre-organized (for the first step of the proline-catalyzed reaction with acetone) 3-MC, 4-MC, and 5-MC (found from conformation search in Spartan and energy-optimized in Gaussian) containing the higher energy conformer (HEC, **1b**) of proline **1**.

**Figure 4 molecules-27-00962-f004:**
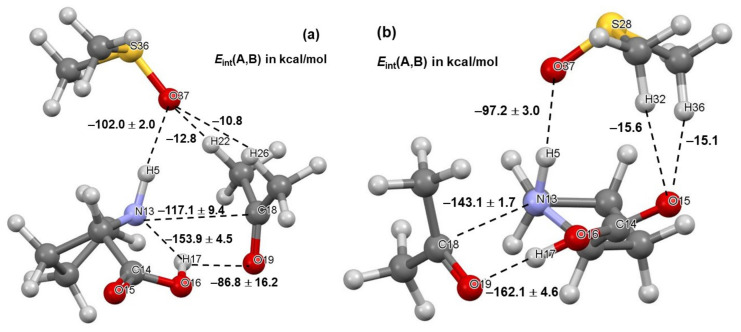
Ball-and-stick representation of 3-molecular complexes involving, besides acetone **2** and a DMSO solvent molecule **3**, the lowest energy conformer (LEC, part (**a**)) and higher energy conformer (HEC, part (**b**)) of proline **1**. The diatomic interaction energies with standard deviations represent average values obtained for 3-, 4-, and 5-molecular complexes with, respectively, one, two, and three explicit DMSO molecules.

**Figure 5 molecules-27-00962-f005:**
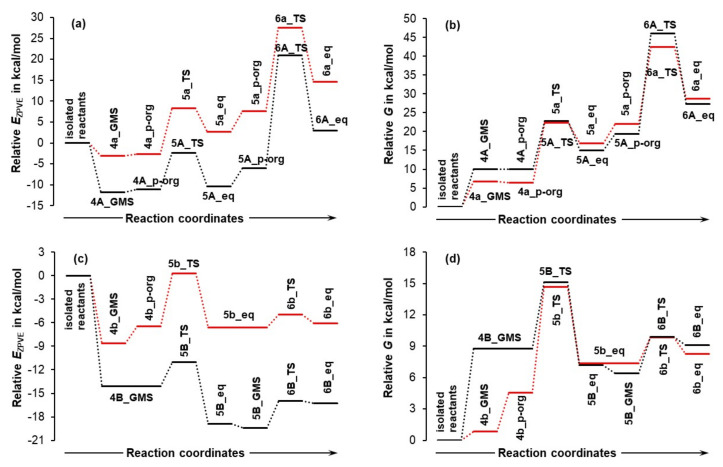
Reaction energy profiles for *E*_ZPVE_ (parts (**a**,**c**)) and Gibbs free energy (*G*, parts (**b**,**d**)) relative to isolated reactants, either (**1**+**2**, for the implicit solvent model) or (**1**+**2**+**3**, in the presence of an explicit solvent molecule of DMSO), reaction energy profiles for *E*_ZPVE_ (parts (**a**,**c**)), and Gibbs free energy (*G*, parts (**b**,**d**)). Data up to the second proton transfer **6a**/**6b** and **6A**/**6B** are presented. The suffix p-org, TS, and eq represent pre-organized, transition state, and equilibrium structures, respectively.

**Figure 6 molecules-27-00962-f006:**
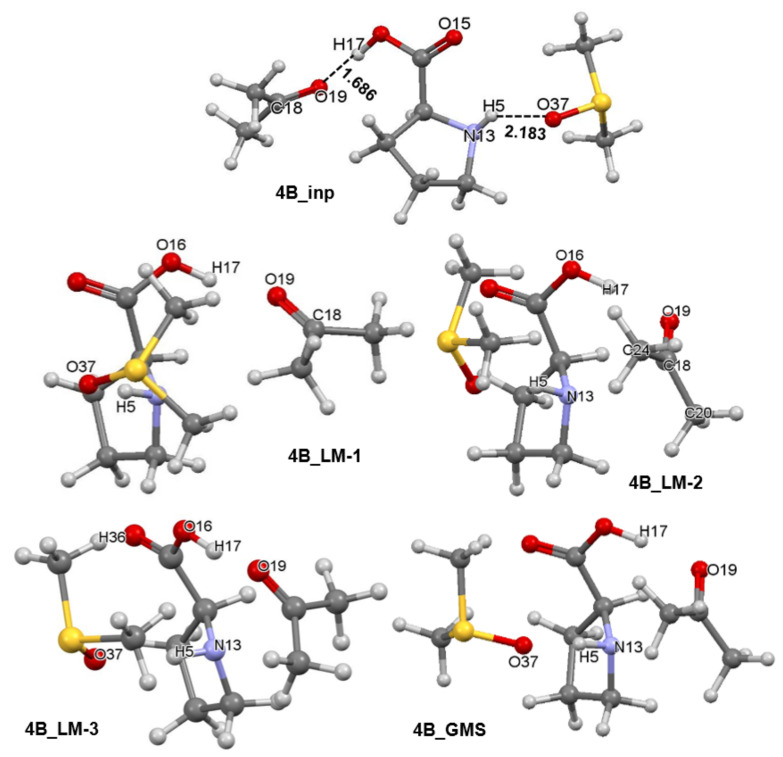
Ball-and-stick representation of 3-MCs showing the initial structure (**4B_inp**), intermediate three local minimum structures (**4B_LM**), and the global minimum energy 3-molecular complex discovered (**4B_GMS**). They all consist of the HEC of proline **1b**, a molecule of acetone **2**, and an explicit DMSO solvent molecule **3**.

**Figure 7 molecules-27-00962-f007:**
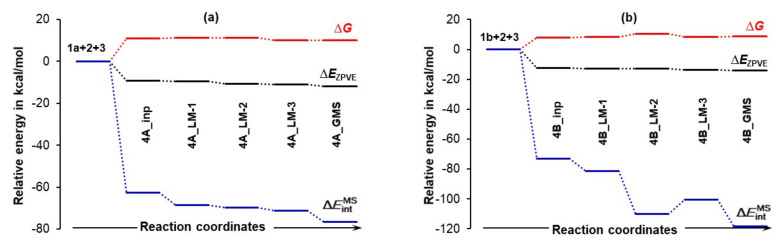
Relative to the energy of separate molecules of **1** (proline, either **1a** in part (**a**) or **1b** in part (**b**)), **2** (acetone), and **3** (DMSO solvent molecule), and energy changes (Δ*E*_ZPVE_, Δ*G* and $\Delta E_{\mathrm{int}}^{\mathrm{MS}}$) computed at the 6-311++G(d,p)/GD3 level in DMSO for the 3-MCs containing **1a** shown in Appendix A and **1b** shown in Figure 6.

**Figure 8 molecules-27-00962-f008:**
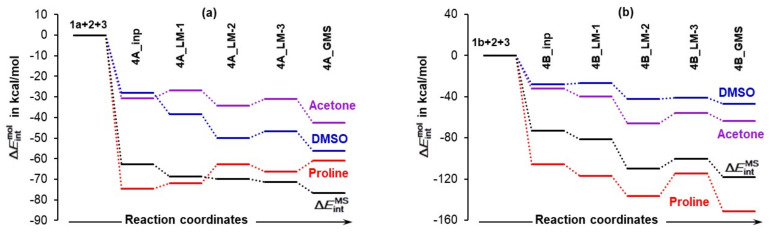
Relative to interaction energies computed for separate molecules, changes, $\Delta E_{\mathrm{int}}^{\mathrm{mol}}$, in the sum of intra- and intermolecular interaction energies computed for the indicated individual molecules constituting a molecular system (either **1a**+**2**+**3** (**4A**) in part (**a**) or **1b**+**2**+**3** (**4B**) in part (**b**)) undergoing a structural change. Inp, LM, and GMS stands for input, local minimum, and global minimum structures of 3-MCs. A trend in $\Delta E_{\mathrm{int}}^{\mathrm{MS}}$ (it accounts for all interactions in a molecular system) is also shown.

**Figure 9 molecules-27-00962-f009:**
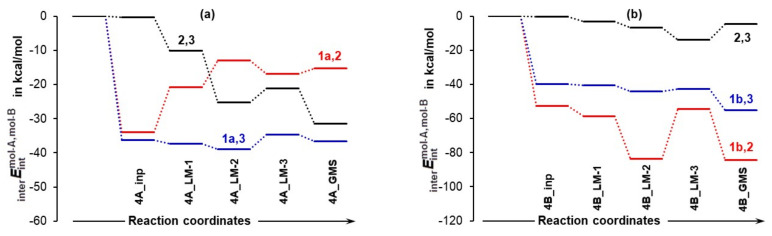
Relative to isolated molecules, the change in the intermolecular interaction energy between molecule-pairs involving proline and acetone **1**,**2**, proline and DMSO **1**,**3**, and acetone and DMSO **2**,**3** in the indicated 3-MCs containing the LEC of proline in part (**a**) and the HEC of proline in part (**b**).

**Figure 10 molecules-27-00962-f010:**
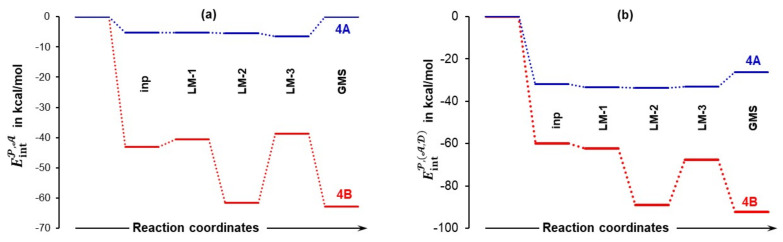
Trends in the interaction energies between molecular fragments: 𝒫 and 𝒜 in part (**a**) and 𝒫 and (𝒜 plus 𝒟) in part (**b**). They were computed for the indicated **4A** and **4B** complexes.

**Figure 11 molecules-27-00962-f011:**
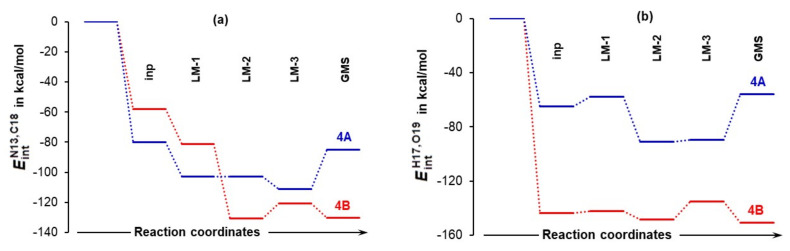
Intermolecular interaction energies computed for the N13,C18 (part (**a**)) and H17,O19 (part (**b**)) atom-pairs in the indicated 3-MCs.

**Figure 12 molecules-27-00962-f012:**
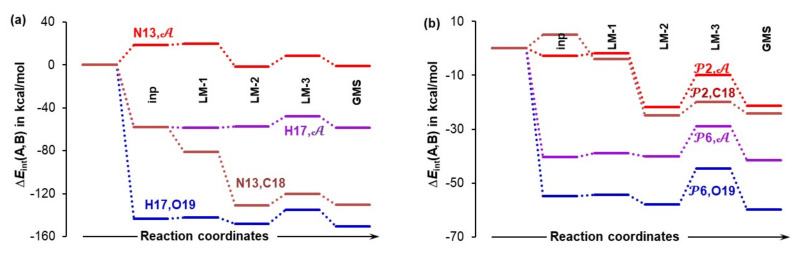
Trends (from an input to the global minimum structure, GMS) in the interaction energy computed for either atom-pairs or atom-molecular fragment pairs: N13,𝒜; H17,𝒜; N13,C18; H17,O19 in part (**a**) and 𝒫2,𝒜; 𝒫2,C18; 𝒫6,𝒜; 𝒫6,O19; in part (**b**). Molecular fragments 𝒫2 = {C1,C4,H5,N13} of the HEC of proline **1b**, 𝒫6 = {C14,O15,O16,H17} of **1b**, and 𝒜 = {C18,O19} of acetone **2** are in **4B** 3-MCs.

**Figure 13 molecules-27-00962-f013:**
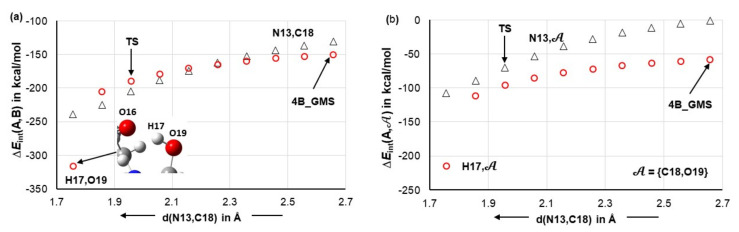
Trends in the interaction energy computed for the N13,C18 and H17,O19 atom-pairs (part (**a**)) and N13,𝒜 and H17,𝒜 atom-molecular fragment pairs (part (**b**)). They were obtained on simulated CN-bond formation by scanning d(C18,13) from the value observed in the **4B_GMS** 3-MC.

**Table 1 molecules-27-00962-t001:** Strongest diatomic attractive (part (a)) and repulsive (part (b)) interaction energies between atoms of the higher energy conformer of proline **1b** and a molecule of acetone **2** in 3-, 4-, and 5-molecular complexes.

**Part (a)**					
**Atoms in**		**Di-Atomic Interaction Energies in kcal/mol**			
**1b**	**2**	**3-MC**	**4-MC**	**5-MC**	**Average**
C14	O19	−188.5	−178.1	−190.1	−185.6 ± 7
H17	O19	−164.8	−156.8	−164.6	−162.1 ± 5
N13	C18	−145.1	−142.1	−147.4	−144.9 ± 3
O16	C18	−135.0	−131.7	−132.0	−132.9 ± 2
O15	C18	−88.8	−85.0	−87.1	−87.0 ± 2
H5	O19	−54.5	−58.4	−57.9	−57.0 ± 2
**Part (b)**					
**Atoms in**		**Di-Atomic Interaction Energies in kcal/mol**			
**1b**	**2**	**3-MC**	**4-MC**	**5-MC**	**Average**
H5	C18	46.7	47.6	48.7	47.7 ± 1
H17	C18	100.2	98.7	97.6	98.8 ± 1
O15	O19	115.4	110.2	116.1	113.9 ± 3
C14	C18	142.5	135.6	140.1	139.4 ± 4
N13	O19	145.7	149.5	147.7	147.6 ± 2
O16	O19	174.4	171.9	176.0	174.1 ± 2

**Table 2 molecules-27-00962-t002:** Energies (in au at the RB3LYP/6-31+G(d,p)/GD3 level in DMSO) computed for the first stage of the catalytic process [23], i.e., the concurrent H-transfer from **1** to **2** and CN-bond formation between **1** and **2**. The lowest energy pre-organized 3-MC, 4-MC, and 5-MC with **1a** (part (**a**)) and **1b** (part (**b**)) were used as input structures for computational modelling. Relative to the energy of the relevant input molecular complex, energy changes (Δ) for a transition state (TS) and equilibrium product (EQ) of the first stage are reported in kcal/mol.

**Part (a)**								
	** *E* **	**∆*E***	** *E* _ZPVE_ **	**Δ*E*_ZPVE_**	** *H* **	**∆*H***	** *G* **	**∆*G***
**3-MCs**								
input	−1147.6522	0.0	−1147.3418	0.0	−1147.3193	0.0	−1147.3962	0.0
TS	−1147.6413	6.8	−1147.3295	7.8	−1147.3093	6.3	−1147.3786	11.1
EQ	−1147.6595	−4.6	−1147.3430	−0.7	−1147.3232	−2.5	−1147.3902	3.8
**4-MCs**								
input	−1700.8866	0.0	−1700.4953	0.0	−1700.4654	0.0	−1700.5614	0.0
TS	−1700.8767	6.2	−1700.4841	7.0	−1700.4565	5.6	−1700.5451	10.3
EQ	−1700.8940	−4.6	−1700.4968	−0.9	−1700.4695	−2.6	−1700.5557	3.6
**5-MCs**								
input	−2254.1254	0.0	−2253.6530	0.0	−2253.6158	0.0	−2253.7275	0.0
TS	−2254.1111	9.0	−2253.6371	10.0	−2253.6026	8.3	−2253.7051	14.0
EQ	−2254.1271	−1.1	−2253.6491	2.5	−2253.6143	0.9	−2253.7190	5.3
**Part (b)**								
	** *E* **	**∆**	** *E* _ZPVE_ **	**Δ**	** *H* **	**∆**	** *G* **	**∆**
**3-MCs**								
input	−1147.6474	0.0	−1147.3360	0.0	−1147.3140	0.0	−1147.3880	0.0
TS	−1147.6450	1.5	−1147.3325	2.2	−1147.3130	0.9	−1147.3800	5.0
EQ	−1147.6613	−8.7	−1147.3454	−5.9	−1147.3260	−7.3	−1147.3930	−3.0
**4-MCs**								
input	−1700.8798	0.0	−1700.4884	0.0	−1700.4580	0.0	−1700.5530	0.0
TS	−1700.8798	0.0	−1700.4868	1.0	−1700.4590	−0.6	−1700.5460	4.6
EQ	−1700.8969	−10.7	−1700.4999	−7.3	−1700.4730	−8.9	−1700.5590	−3.4
**5-MCs**								
input	−2254.1236	0.0	−2253.6494	0.00	−2253.6130	0.0	−2253.7210	0.0
TS	−2254.1223	0.8	−2253.6471	1.5	−2253.6130	0.3	−2253.7140	4.0
EQ	−2254.1362	−7.9	−2253.6577	−5.2	−2253.6240	−6.5	−2253.7240	−2.1

## Data Availability

On request, computational data are available from I.C.

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
