# Peer review of "A Molecular-Wide and Electron Density-Based Approach in Exploring Chemical Reactivity and Explicit Dimethyl Sulfoxide (DMSO) Solvent Molecule Effects in the Proline Catalyzed Aldol Reaction"

_molecules, 2022, doi:10.3390/molecules27030962_

Round 1
Reviewer 1 Report
Report on the paper entitled "A molecular-wide and electron density based approach in exploring chemical reactivity and explicit DMSO solvent molecule effects in the proline catalysed aldol reaction. ", by Ignacy Cukrowski, George Dhimba and Darren L. Riley, submitted to Molecules.
The present paper explores the role of one DMSO molecule in the (DMSO solvated) proline catalysed aldol reaction. For the purpose they used the recently published (by the same authors) approach named REP-FAMSEC (reaction energy profile-fragment attributed molecular system energy change). The present work underlines the powerfulness of the protocol to evidence the interactions leading to the mechanism. They used the Molecular-Wide and Electron Density based concept of Chemical Bonding (MOWED-CB) recently promoted by one of the authors to determine the driving forces in the formation of chemical bonds.
Interestingly, they summarised the basic concepts used, so that the reader does not necessarily need to go the references in order to understand the method. (Indeed the referee did it, in order to verify if a similarity/plagiat was not present).
The paper is well written, including up to date references, and describes satisfactorily the underlying theory.
I recommend publication in the present form.
Author Response
We are very happy with the reviewer’s comments and appreciation of contribution made and its significance. Indeed, we summarised basic concepts of the REP-FAMSEC method developed by us recently. This was done on purpose, as we wanted to make it easier for a reader to follow our new developments and protocols implemented in this work. In fact, to make it even more convenient, we prepared an Appendix A that contains uncommon in the field symbols and their meaning.
Reviewer 2 Report
The Authors have approach to the modelling of the proline catalysed aldol reaction the presence of an explicit molecule of DMSO which has showed that DMSO is a major player in the aldol reaction as it plays a double role. The manuscript deserves to publish in Molecules after a major correction. I would like to suggest introducing changes before publishing in Molecules.
The authors should revise in the manuscript as the following points:
- The title and abstract should not contain abbreviations. If they do appear, they should be explained. Please correct the abstract and the title in this respect.
- Why the Authors used RB3LYP, M06 and M06-2X methods? Whether another method was taken into account? Are there any links to the experimental data?
- In Table 2, please add symbols ΔE, ΔG, ΔH. In addition, please insert calculations for ΔS
- The article contains minor linguistic errors. Please check your spelling again
- Please complete conclusions to combine theoretical work with future or proven experimental work.
Author Response
Comments and Suggestions for Authors
The title and abstract should not contain abbreviations. If they do appear, they should be explained. Please correct the abstract and the title in this respect
Our reply:
Thank you for your suggestion. The abbreviation DMSO has been now explained in the title and the abstract.
============================================================================
Why the Authors used RB3LYP, M06 and M06-2X methods? Whether another method was taken into account? Are there any links to the experimental data?
Our reply:
Note that this is a theoretical work and our computational modelling was done at the RB3LYP/6-311++G(d,p) level of theory. The selection of the level of theory is fully explained in the 2nd section , i.e., 2. Computational details. It reads:
Starting from line 86
‘All calculations were performed in Gaussian 09 Rev. D01 [35] at the RB3LYP/6–311++G(d,p) level of theory with Grimme’s [36] empirical correction for dispersion (GD3). It was established that, using CCSD(T) as a benchmark, B3LYP-GD3, M06 and M06-2X (with the 6-31+G(d,p) basis set) give a reasonable, within few kcal/mol, electronic and Gibbs free energies in modelling reaction mechanisms [37].’
This means that these three approximations could, in principle, be used to reproduce electronic and Gibbs free energies computed at the very expensive CCSD(T) level of theory. However, we opted for the most frequently used in the field B3LYP. This is because one of the authors (IC) has tested reliability of IQA data generated from the B3LYP-generated wavefunction file. To this effect, see the text starting from line 98:
‘Topological calculations were performed in AIMAll (ver. 19.02.13) [38] using B3LYP-generated wavefunctions as IQA energy terms, and interaction energies in particular, were found to be highly comparable to those obtained at the CCSD/BBC1 level [39].’
The reviewer (and a reader) can also consult a paper that explains superior quality of the CCSD/BBC1 IQA data relative to commonly available (in AIMAll) CCSD/Müller IQA data (I. Cukrowski and P.M. Polestshuk, “Reliability of Interacting Quantum Atoms (IQA) Data Computed from Post-HF Densities: Impact of Approximation Used”, Phys. Chem. Chem. Phys. 19 (2017) 16375 16386. DOI: 10.1039/c7cp02216f.
Throughout our paper, references are made to the experimental work done by other researchers.
===========================================================================
In Table 2, please add symbols ΔE, ΔG, ΔH. In addition, please insert calculations for ΔS
Our reply:
Thanks, it has been done except ΔS values. Our novel approach is focused mainly (but not exclusively) on the interaction energies as driving forces for a chemical change. We do not discuss, not even mention, entropy in our manuscript and not to confuse the reader, we decided not to include these values in Table 2.
============================================================================
The article contains minor linguistic errors. Please check your spelling again
Our reply:
Thanks for this suggestion. We did our best and relevant changes made are seen in the revised version.
============================================================================
Please complete conclusions to combine theoretical work with future or proven experimental work.
Our reply:
The proven experimental work is already mentioned – see line 1076.
Round 2
Reviewer 2 Report
The authors correctly answered to the questions and the manuscript can be published in the present form.